# Intelligent Approach to Network Device Migration Planning towards Software-Defined IPv6 Networks [note 1]

**DOI:** 10.3390/s22010143

**Published:** 2021-12-26

**Authors:** Babu R. Dawadi, Danda B. Rawat, Shashidhar R. Joshi, Pietro Manzoni

**Affiliations:** 1Department of Electronics and Computer Engineering, Pulchowk Campus, Tribhuvan University, Kathmandu 19758, Nepal; srjoshi@ioe.edu.np; 2Cyber Security and Wireless Networking Innovations Lab, EECS Department, Howard University, Washington, DC 20059, USA; db.rawat@ieee.org; 3Department of Computer Engineering, Universitat Politècnica De València, 46022 Valencia, Spain

**Keywords:** ANFIS, SDN, IPv6, SoDIP6, network device, migration planning

## Abstract

Internet and telecom service providers worldwide are facing financial sustainability issues in migrating their existing legacy IPv4 networking system due to backward compatibility issues with the latest generation networking paradigms viz. Internet protocol version 6 (IPv6) and software-defined networking (SDN). Bench marking of existing networking devices is required to identify their status whether the existing running devices are upgradable or need replacement to make them operable with SDN and IPv6 networking so that internet and telecom service providers can properly plan their network migration to optimize capital and operational expenditures for future sustainability. In this paper, we implement “adaptive neuro fuzzy inference system (ANFIS)”, a well-known intelligent approach for network device status identification to classify whether a network device is upgradable or requires replacement. Similarly, we establish a knowledge base (KB) system to store the information of device internetwork operating system (IoS)/firmware version, its SDN, and IPv6 support with end-of-life and end-of-support. For input to ANFIS, device performance metrics such as average CPU utilization, throughput, and memory capacity are retrieved and mapped with data from KB. We run the experiment with other well-known classification methods, for example, support vector machine (SVM), fine tree, and liner regression to compare performance results with ANFIS. The comparative results show that the ANFIS-based classification approach is more accurate and optimal than other methods. For service providers with a large number of network devices, this approach assists them to properly classify the device and make a decision for the smooth transitioning to SDN-enabled IPv6 networks.

## 1. Introduction

The world’s information and communication technology (ICT) industries are moving to the very first new technologies like the 5G-based mobile network [1,2], industry 4.0 with the vision for industry 5.0-based society [3,4,5], cloud services with the advancement on modern datacenter operation and management, Internet protocol version 6 (IPv6) addressing mechanism [6], software-defined networking (SDN) [7] paradigm, and many more. This is all about the nature of human beings who always seek to have fast, efficient, and reliable services in their daily lives. There are many challenges to transform the existing internet infrastructure to newer technologies. For example, higher costs of investment, lack of stable applications and protocol standards, security and quality of service issues, skilled human resources management and many more [8,9]. However, suitable solutions for smooth transitioning of existing networks into new networking paradigms like SDN and IPv6 are undergoing researches [10,11,12]. In terms of new technology implementations in operational networks, the invention of newer technologies and their application in practice causes obstacles for service providers. The existing networking infrastructure providing older technology-based services are to be transformed to newer infrastructure and services. But, the immediate service provisioning with invented newer technologies is not possible due to the higher cost of investment for service providers [9].

A device upgrade is preferable than a replacement in order to save greater capital and operational expenditures (CapEX and OpEX). Because, when considering networking infrastructure management, upgrade costs are often cheaper than replacement costs. However, all networking devices could not be upgradable to newer technologies at a time. Additionally, based on the interrelationship between IPv6 and SDN, joint migration decision being taken by service providers towards SDN and IPv6 networks migration is more cost effective than individual migration implementation [13,14]. The major steps for service providers for joint network migration to SDN and IPv6 are:Assess the device performance and status towards the possibility of their hardware/firmware upgrades or replace the equipment with a newer one to make it operable with newer networking paradigms viz. SDN and IPv6.Identify the total cost of network migration for upgrades or replacement.Develop the strategic and sustainable migration plan in a phase by considering budget constraints, human resource requirements, technology readiness, business continuity planning, and many more.Implement the migration and foresee new business opportunities.

In this paper, we focus on a solution to assess the existing networking devices for efficient transformations of existing network infrastructure into SDN-enabled IPv6 network termed software-defined IPv6 (SoDIP6) network [8] with optimum costs. This ensures future sustainability of service providers against the higher cost of investment. Considering the network migration planning, the major question in the beginning is—“are the existing networking infrastructure operating with older technologies migrate-able to operate with newer technologies?” SDN and IPv6 networking paradigms, unfortunately, are not backward compatible. As a result, older networking equipment must either be replaced or have their hardware/software changed while in use in order to provide newer technologies and services.

To the best of our knowledge, we find many researchers suggesting for the phase-wise or incremental deployment of SDN and IPv6 networks for smooth transitioning, which we will highlight in Section 2 as related work, but we could not find the research that addresses the question we raised above. To answer the preceding issue, we must first assess the device condition and then recommend the decision maker on how to proceed with network migration planning. Many classification algorithms in machine learning are available for which based on the data patterns available and design of input variables, we comparatively present the approaches and evaluate the performance using regression tree, support vector machine, ensemble tree, and implement adaptive neuro fuzzy inference system (ANFIS) to solve this migration problem.

ANFIS is a well-known intelligent approach applicable to solve these problems, which are particularly suitable for classification, estimation, prediction, and forecasting. For network device status identification, whether it is upgradable or replaceable, in this article, we apply ANFIS, the combination of artificial neural network and fuzzy inference system. We model ANFIS based on its appropriateness for mathematical analysis and greater computing efficiency, following Takagi–Sugeno fuzzy rules [15]. The identification of a set of input parameters for ANFIS, particularly of a network device, is a complex problem as we need to deal with the data of both qualitative and quantitative types. After a running series of steps in preprocessing to prepare the dataset, input dataset for dependency fuzzy system (DFS) and ANFIS will be developed for training, testing, and validation. The preliminary analysis version of this work was presented at CCNC-2021 conference [16], while in this article, we performed the detailed and extended analysis by considering wide range of dataset with mathematical formulations, improvement on proposed implementation framework and algorithms, and performance evaluation as well as comparisons with other different recent classification methods. The major contributions of this paper are as follows.

ANFIS-based classification approach is proposed to identify the network device (IP router) status for an upgrade or replacement towards transition to SoDIP6 networks.The proposed model is compared with other classification models in which our proposed model presented a better accuracy.We implemented our proposed approach over two standard IP routing topological networks (UNINET and CERNET) and achieved accurate results.This approach is contributory to Internet and telecom service providers having a large number of network devices to be considered for migration with sustainable migration planning of their existing legacy IPv4 networks into SoDIP6 networks.

The rest of this paper is organized as follows. Section 2 presents the background of our research with related works on ANFIS modeling and implementation. We will discuss our proposed approach with the implementation framework and data pre-processing, training, testing, and validation in Section 3. The proposed model will be implemented, and finally evaluate the model with analysis and discussion in Section 4, while Section 5 concludes the paper.

## 2. Background and Related Work

The SDN paradigm is expected to be the most featured solution for network and Internet service providers as well as cloud service providers worldwide in terms of ease of network operation and management with optimized OpEX. Similarly, IPv6 addressing avoids all the routing and associated issues (e.g., NAT proliferation, address auto-configuration, oversized packet fragmentation, hierarchical routing management, security and quality of service, etc.) with address depletion problems in legacy IPv4 networking. Hence, SDN and IPv6 networks, jointly known as the Software-defined IPv6 (SoDIP6) network [8] is regarded as the most efficient latest generation networking paradigm to be adopted by stakeholders like enterprises, telecom, Internet, and cloud service providers worldwide [17] for future sustainability and competing with global technology changes.

Several transition mechanisms have been devised and under implementations [18,19] for migrating existing IPv4 networks into an operable IPv6 networking system. SDN migration procedures have also been outlined, with ISPs being able to choose their own migration strategies based on their present network circumstances [12]. Migration to SDN in telecom operators’ (Telcos) and Internet service providers’ (ISP) networks have evolved with significant progress on developing transition technologies [12,20,21]. In this regard, based on the inter-relationship defined between SDN and IPv6 networking paradigms, we have developed the proper transition planning and migration cost minimization with their benefits and challenges of migration for joint migration to the SoDIP6 network in our previous works [9,13,17,22,23,24].

Service providers could have hundreds of thousands of switches/routers running in their network that will not be able to migrate those networking devices at once. Additionally, the major concern is that each service provider has to confirm with their network devices whether they are upgradeable or should be replaced to make them operable with newer technologies and applications associated with SDN and IPv6. Small and medium enterprises (SMEs), and service providers of developing countries run their network devices even after the device end-of-support (EoS) due to a higher cost of investment. Considering the network migration steps presented in our previous work [17], an intelligent approach to identify the status of network devices is to be developed before addressing the decision of migration so as to have proper planning and management of budget and human resources required for the migration.

In the process of migration planning, service providers have to be confident with respect to the following questions regarding their network devices.

Is the device Internetwork operating system (IoS)/firmware upgradable to enable operation of IPv6 and SDN?Is the existing memory and processing capacity of the network device sufficient to operate with newer technologies if they are upgraded or does it have an extra slot for memory/processing capacity addition?What is the end-of-life (EoL) announcement date of the device? How many years does it have to operate?What is the end-of-support (EoS) date of the device? Is the vendor ready to provide support for next couple of years?What is the device throughput? Is it sufficient to operate with upgraded newer technologies and applications?

We can identify the set of input parameters to be considered for device status identification from the above questions. Major parameters we consider are: (a) Upgrade on IPv6 and SDN-enabled IoS/Firmware—binary value (True/False), (b) storage capacity—quantitative value (MB), (c) device throughput—quantitative value (Mbps), (d) EoL—quantitative value (years), and (e) EoS—quantitative value (years).

The majority of parameter values may be derived from technical specifications that will be stored in the knowledge base (KB) system, while others, such as the average of maximum CPU usage, memory utilization, and throughput, can be extracted in real time using an SNMP agent. Hence, using the KB, set of input data will be prepared and then those parameter values will be input to ANFIS to classify the device for migration planning.

### 2.1. Overview of Adaptive Neuro Fuzzy Inference System (ANFIS)

The adaptive network-based fuzzy inference system [25] is also known as a hybrid neuro-fuzzy system technique. It is made up of two machine learning techniques: Artificial neural networks (ANN) and fuzzy inference systems (FIS). The neuro-fuzzy inference system has been found to be a strong computational model for classification, estimation, and forecasting in a variety of domains. It resembles a feedforward neural network with each layer representing a neuro-fuzzy component system.

In ANFIS, fuzzy logic accounts for system imprecision and uncertainties, while the neural network provides flexibility. ANFIS creates a fuzzy rule-base first, then uses the trained dataset to tweak the parameters of the membership functions [26]. The ANFIS is primarily organized as a five-layered system. Input, if part, rules and normalization, then part, and output make up the five levels. It could have distinct nodes in each layer connected to nodes from the previous level, with the output of the previous level serving as the input signals for the subsequent layer. For example, for the Takagi–Sugino rules type [27], the typical common rules with two input and one output variables in the model as depicted in Figure 1 can be determined as follows.

if x is $A_{i}$ and y is $B_{i}$ then $f_{i}=p_{i}\cdot x+q_{i}\cdot y+r_{i}$                      rule-i

Hence, the possible rules are:

if x is $A_{1}$ and y is $B_{1}$ then $f_{1}=p_{1}\cdot x+q_{1}\cdot y+r_{1}$                     rule-1

if x is $A_{1}$ and y is $B_{2}$ then $f_{2}=p_{2}\cdot x+q_{2}\cdot y+r_{2}$                     rule-2 

The first layer defines the membership function for each *i*th node. The fuzzification of the input variables are performed with the output shown in Equation (Equation 1):(1)$$O_{i}^{1}=\mu_{A_{i}\left(x\right)}.$$

$O_{i}^{1}$ is the output of the *i*th node and is the membership grade of a fuzzy set (A1, B1), where (A1, B1) represents the linguistic level associated with node *i*.

The layer 2 nodes are the fixed node that represents the firing strength of the rule and consists of the product (AND) of the antecedent part of the fuzzy rules (incoming signals) as shown in Equation (Equation 2):(2)$$O_{i}^{2}=w_{i}=\mu_{A_{i}\left(x\right)}\cdot \mu_{B_{i}\left(y\right)},i=1,2.$$

Similarly, the output of the third hidden layer normalizes the membership function and gives the normalized (N) firing strengths. The *i*th node calculates the *i*th rule’s firing strength to the sum of all rules firing strengths is shown in Equation (Equation 4):(3)$$O_{i}^{3}={\bar{w}}_{i}=\frac{w_{i}}{w_{1}+w_{2}},i=1,2.$$

Layer 4 nodes are the adaptive nodes that provide de-fuzzification in which the consequent parameters of the rule are determined with a node function having $p_{i},q_{i},r_{i}$ as the parameter set. Hence, Equation (Equation 4) gives the output of layer 4:(4)$$O_{i}^{4}={\bar{w}}_{i}f_{i}={\bar{w}}_{i}\left(p_{i}x+q_{i}y+r_{i}\right),i=1,2.$$

Layer 5 provides the single node output as shown in Equation (Equation 5). It computes the overall output as the summation of all incoming signals:(5)$$O_{i}^{5}=\lim_{i}\sum {\bar{w}}_{i}f_{i}=\frac{\lim i\sum {\bar{w}}_{i}f_{i}}{\sum {\bar{w}}_{i}}.$$

In ANFIS, premise parameters (to learn the parameters related to membership functions) are determined using the back-propagation learning algorithm, while the least square estimator is used to determine the consequent parameters. The premise and consequent parameters are determined in the training phase using a training dataset, while an error threshold is defined between the actual and desired output. The forward pass and backward pass procedures in ANFIS are used to learn the parameters. The input patterns are transported from input to output using an iterative least mean square approach to estimate the optimal consequent parameters in the forward pass, and the premise parameters are set in the concurrent training cycle. The error signals are back propagated in the backward pass to alter the premise parameters on this epoch while keeping the subsequent parameters fixed. The output converges towards error threshold defined by propagating back the error and update premise parameters using the gradient descent method.

In our proposed model, a knowledge base (KB) system will be established from the device specification and other external sources, while the real time performance parameters for example, processing, memory, and throughput will be collected using the SNMP agent. Hence, five input variables and one output binary variable that provides the device status in the ANFIS structure are defined.

### 2.2. Related Work in ANFIS Implementation and Network Migration

We found limited literature for the incremental deployment of SDN and IPv6 networks with an overall migration plan for ISPs and Telcos. For example, studies like the game theoretic approach on IPv6 network migration [28], incremental adoption to IPv6 networks [29], evolutionary process on SoDIP6 network migration [17], agent-based modeling for joint migration to IEEE-PCE and SDN [30], SoDIP6 network migration based on customer priority, and optimal path [13,22], incremental deployment of hybrid SDN in service provider networks [31,32,33,34] and optimal sequence of router replacement using greedy algorithm for SDN migration [21] are those studies, which provide insights into individual and joint network migration. However, these studies were all concerned with the migration to be implemented only after finding the status of a device whether to migrate or upgrade. We lack the literature, particularly on how service providers assess whether a device should be replaced or updated before considering migration initiatives. Device status detection is also a major concern for regular upgrades and maintenance, since regular patch up of the software/firmware and upgrades on hardware is required to be updated with the latest threats and service quality. In our previous work [35], we implemented ANFIS to identify the network switch status for planning upgrades, particularly focusing on regular upgrades and the maintenance of the ISP network. This encouraged us to implement ANFIS for network device (router) transition planning towards migration to the SoDIP6 network with this study.

To the best of our knowledge, there are few or no research papers on the use of ANFIS in SDN and IPv6 network migration, yet ANFIS has a wide range of multidimensional implementations and applications. We went through some literature of ANFIS implementations in communication networks, at which it is mostly used in estimation, prediction, optimization, and forecasting [36,37,38,39,40,41,42,43,44,45,46,47,48,49,50], but none of these studies are particularly related to SDN and IPv6 network migration.

Kumaravel A. et al. [51] implemented the malicious node detection system in MANET using ANFIS. The author used throughput, average packet loss ratio, energy consumption, and detection ratio as the major parameters for the input to ANFIS to make a classification and performance evaluation of the proposed model.

Mummolo G. et al. [52] introduced ANFIS to classify medical equipment that needed to be replaced. The author first used a scoring method to determine the values of each input parameter. The system then makes a recommendation for medical equipment replacement based on the downtime ratio, maintenance ratio, age ratio, and redundancy ratio. These types of input parameters can also be used to determine the status of an ISP network device as part of a routine maintenance schedule. Our choice of device parameters proposed in this study are more relevant with the technology transformations form legacy to SoDIP6 network capability. The parameters conceptualized at [35,52] are applicable only to the performance upgrade on the existing devices and on the same technology as a part of regular maintenance plan. However, for our case, we only consider migration of network devices in terms of support for new technology e.g., SDN and IPv6 operation.

ANFIS is also popularly used to address classification problems [51,53,54]. Our research problem is also related to the detection of the network device for its upgrade or replacement via ANFIS classification in the domain of new network deployment.

## 3. Proposed Approach

To avoid the possible transition issues with base-line consideration to minimize the organizational CapEX and OpEX for sustainable future societies, ISPs have to migrate their network into a dual-stack SoDIP6 (dual-stack IPv6 and hybrid SDN) network in an incremental basis based on customer priority and optimal path routing [22] so that every router after migration can operate as a legacy stack and SDN stack based on the options available in its IoS. It means, SDN and IPv6 capabilities are simply enabled in the legacy routing gear by embedding separate software component [55] or via IoS upgrades or the legacy gear is to be replaced to make it operable with dual-stack SoDIP6 networks.

In general, every ISP keeps a list of its network’s routing devices with specification data in its network inventory management system and regularly monitors network operations to avoid potential failures and issues during service delivery. Most of the device details can be extracted from the device specifications. The issue is that only a few legacy routing devices support IPv6 addressing, and all operational routers do not support the SDN/OpenFlow protocol [56]. Ultimately, all such routing devices must be replaced or modified to support IPv6 addressing and the OpenFlow protocol. Figure 2 depicts the enhanced version of the overall implementation framework [16] for our suggested strategy. Regular vendor alerts, new technologies and applications, as well as other domain-specific knowledge, are all important sources for keeping network device parameter information in the KB up to date. With the stated IoS/firmware version, KB keeps track of what the current version supports, such as whether it supports IPv6 and OpenFlow, as well as information on the extra memory slot, device EoL, and EoS. The proposed KB provides answers to the following questions.

Is the current IoS/firmware version upgradable?Which new IoS/firmware version does it support?Does the new version support IPv6 and SDN/OpenFlow?Does the network device have an extra memory slot for addition?How many years of life the network device has spanned?How many years is vendor support available?

The defined input parameters and their descriptions are provided in Table 1 for our proposed model. Only if the network device supports updated IoS/firmware capable of IPv6 and SDN/OpenFlow operation can it be upgraded. As a result, as an output from the dependency fuzzy system, the Boolean variable ‘IO’ offers a TRUE and FALSE status. DFS makes the choice on prerequisite characteristics such as IoS/firmware upgrade, additional memory slot availability, new IoS support for IPv6, and SDN. As a result, ANFIS input is dependent on DFS first. Every network device has an operational life beyond that must be replaced, and the specified equipment’s end-of-support period is determined by the vendor support period. IP routers generally have an operational life span of 4–5 years [57], however, the present technology will become obsolete, and the need for effective services within the life span of network equipment has necessitated a fast update or replacement of network devices. Generally, the lifespan depends on the vendor’s quality of production as to be specified by vendor notifications [58,59].

Vendors announce the end-of-support date for their outdated equipment on a regular basis [60]. As a result, a KB keeps the EoL, EoS, and memory capacity of earlier equipment with improved version support. The available information is utilized to determine the device’s upgradeability. Since our goal is to make the device operational with IPv6 and SDN, adding memory or processor capacity without support for IoS/firmware upgrades is pointless. Hence, the IPv6 and OpenFlow support status is fetched into DFS first, which decides the upgradability of device IoS/firmware and hardware. If the system software does not support new version capable of the SoDIP6 network, then a zero value as an output of expandable memory size is calculated. This leads to the replacement decision by ANFIS. The fuzzy rules ((R1)–(R4)) [16] for DFS are defined accordingly as follows.

If *device supports IoS/Firmware upgrade*, then New-Version = Upgraded-Version           (R1)

If *Upgraded-Version is SoDIP6 capable*, then IO = 1                         (R2)

If *Upgraded-Version is not SoDIP6 capable* then IO = 0                       (R3)

If *device has extra memory slot then expandable memory (E)* = $M_{E}$                   (R4)

Based on the output from DFS, additional data of a router will be obtained from the operational network in real time via the simple network management protocol (SNMP) agent and combined into the input dataset to fetch into ANFIS for status identification. The remaining lifespan and support period in years, as well as the total unused memory including expandable memory, are calculated as follows to generate the dataset.

L = (Current Date − EoL Date),

S = (EoS Date − Current Date),

M = (m + E) · IO,

The variable ‘m’ represents the average amount of unused memory space in the device while it is in use. Based on the EoL announcement date, we calculate EoL in years. Six months before the effective date of EoL adoption, hardware vendors, such as Cisco issue an EoL notice. Only vendor supports are provided after the EoL announcement date for 5 years till EoS date. The device needs to be replaced after EoS.

IP routers have flash memory to run IoS/firmware and DRAM memory for other processing, for example, packet buffering, maintaining the routing table, security, and QoS implementations. For faster operation, it requires a bigger size IPv6 packet for forwarding and maintaining a larger-sized flow table after upgrades, which most importantly require a higher-sized DRAM. In this article, we consider the expansion of DRAM as one parameter to consider for device migration.

We consider the transformations of only legacy devices in the beginning stage. However in the consequent stages, where some devices are already transformed after applying this proposed approach, then the DFS provides a solution to “Do nothing” if the device IoS is already SoDIP6 capable. This avoids the additional processing burden for those devices that are already SoDIP6 capable. If the device does not support IoS/firmware upgrades or the upgraded version does not support IPv6 and SDN, then the only solution is to purchase a SoDIP6 capable device and replace it. The system is dependent on the software upgrade to proceed further for identification. Hence, a dependency fuzzy system is implemented before processing to ANFIS. The DFS module determines the system’s upgradeability first. However, the system’s complete update is not solely dependent on the software upgrade chosen by DFS. For the final prediction of device upgradeability, the device’s total processing capacity, memory adequacy, and throughput are also taken into account. The SNMP agent collects data about memory, throughput, and processing capacity in real time. DFS will obtain information from KB to determine the relevant data of IO, L, S, and M using device mapping. Similarly, other parameters e.g., average CPU usage, memory unused, and throughput will be obtained from a real network operation using SNMP.

Algorithm 1 presents the steps to implement DFS and ANFIS to get the device status [16]. Function ‘DFS()’ identify the parameters ‘L’, ‘S’, and ‘$M_{E}$’ for a router using KB. This function also provide the solution that if the device IoS is already SoDIP6 capable, then further execution of the algorithm to identify the device status is not required. Similarly, unused memory (m), ‘T’, and ‘C’ are obtained from the SNMP agent. A middleware API is developed to call the trained ANFIS model with defined input parameter values to identify the status.
**Algorithm 1:** ANFIS implementation for network device status identification.
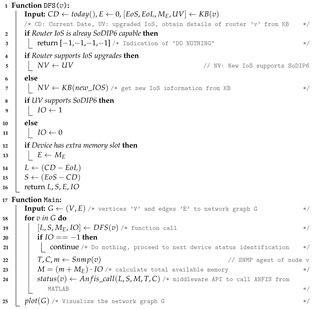


### Dataset for Training, Testing, and Validation

We consider dataset generated from 40 different CISCO product models referring to CISCO 800, 1700, 1800, 1900, 2600, and 2900 series routers. CISCO products are popularly used worldwide in service provider networks. Compared to other vendor products, CISCO products come with a wealth of specification details, as well as important notices and information. CISCO also communicates the EoL and EoS of its products with its authorized dealers on a regular basis worldwide [59]. Based on popularity, we refer to the CISCO routers to test and validate our proposed model. Taking the reference from CISCO IP routers, more than 900 data samples were generated randomly for testing and validation, via augmentation by using the minimum and maximum value of each parameter of the train dataset as confidence intervals. Almost 60% of the dataset were used for training, 20% for testing, and 20% for validation. Most of the router models used in this evaluation support IPv6, but does not support OpenFlow. We assume that the upgraded version of the IoS release fully supports IPv6 and SDN/OpenFLow. As a result, the DFS module decides whether to proceed with ANFIS input based on the device’s new IoS version’s IPv6 and OpenFlow capability. Note that the term ‘SDN’ and ‘OpenFlow’ are used interchangeably in this article. Since, OpenFlow is the southbound API of SDN to enable communication with data plane devices.

Input parameters are mapped to the scoring system indicated in Table 2 for ease of model operation and to reduce the error margin. As part of data refining, EoS is given a higher weight based on sensitivity, as generally most of the CISCO IP routers have expandable memory slots, therefore memory is given a lower weight value. Lower the overall score value has higher significance to replace the device, while a higher score value is supportive of an upgrade. Based on the maximum and minimum value of the trained dataset, the range of value is defined in our scoring system.

We captured the device specification details including IoS release versions, IoS upgrade history of the device, and SDN and IPv6 support with the parameters defined in Table 1. Simulation was run to obtain the average minimum RAM unused, average maximum throughput, and average maximum CPU utilization from a real-time operational network. The distribution of data samples (considering 500 samples) for each parameter plotted in the range from minimum to maximum is shown in Figure 3. It visualizes the range of data samples used in this experimental analysis. The lifespan (L) of device ranges from −2 to 14 indicates that some devices are running beyond the standard life of the operation. Similarly, EoL period ranges from −10 to 5 years indicates that devices are running beyond the end of support. Memory of devices ranges up to 2 GB, throughput (T) ranges up to 2 Gbps, and CPU utilization (C) varies from 10% to 90%.

Individual input variable related to upgrade or replacement is interpreted as shown in Figure 4. A score greater than ‘2’ for each input parameter is supposed to be appropriate for upgrades. The device operation life span shown in Figure 4a is 5 years, while Figure 4b–d have their usual interpretations. Figure 4e provides the score mapping range based on the input data.

The modeled system is theoretically understood as given in Equation (Equation 6) [16] if ‘δ’ is the tuple corresponding to any or all input variables and *Z* is the output variable.
(6)$$\forall \delta \in \left[L,S,M,T,C\right],Z=ANFIS\left(\delta \right)=\left\{\begin{array}{c} \leq 0\;\left(\mathrm{Replace}\right),\delta =-16\\ >\;2\;(\mathrm{Upgrade}),\;\delta \in [2,3,4]\\ \leq 2\;(\mathrm{Replace}),\;\delta \in [2,3,4] \end{array}\right.$$
(7)$$\forall \delta \in \left[L,S,M,T,C\right],Z=ANFIS\left(\delta \right)=\left\{\begin{array}{c} 0\;\left(\mathrm{Replace}\right),\\ 1\;(\mathrm{Upgrade}),\; \end{array}\right.For\;\delta \in \left[-16,2,3,4\right]$$
(8)$$\forall \delta \in \left[L,S,M,T,C\right],Z=ANFIS\left(\delta \right)=\left\{\begin{array}{c} \leq 0.5\;\left(\mathrm{Replace}\right),\\ >0.5\;(\mathrm{Upgrade}), \end{array}\right.for\;\delta \in \left[-16,2,3,4\right]$$

Equation (Equation 6) provides the result based on the weight value assigned on each input variable with the output (*Z*) interpreted by Equation (Equation 9):(9)Z=w1·L+w2·S+w3·M+w4·T+w5·C.

In the worst case, if all input variables have a minimum value of −16 or maximum value of 4, then using Equation (Equation 6) provides the output ranging from −16 to 4. However real data have an output classification field of either 0 or 1 to map the result as either replace (0) or upgrade (1). Hence, ANFIS output is interpreted by Equation (Equation 7), while if the fuzzy output is considered, we consider 50% of the highest weighted average score as the threshold defined by Equation (Equation 8) [35] and interpret the output weight factor given by Equation (Equation 9). The output of ANFIS is a fuzzy value in which we can not avoid the error tolerance. Hence, the prediction is alternatively provided based on the threshold defined by Equation (Equation 6).

## 4. Experimental Analysis and Evaluations

### 4.1. Training and Evaluations

In this section, we run the experiment using the generated dataset to train the ANFIS model, testing, and validation. We suppose that ISPs maintain the inventory systems of their network devices with KB for each device in the network that are in operations. Once the model is trained, ISPs can run the model any time or routinely every 6 months of regular maintenance period to get the status of their devices. Based on detection, they can plan for migration and budget estimation for their legacy network migration to the SoDIP6 network.

We used the MATLAB fuzzy logic toolbox to train the model, utilizing a hybrid learning approach with distinct membership functions (MF) and types trained at various epochs. At first, the model was trained with a primarily refined dataset before applying scoring and achieved average testing RMSE of 0.7503 at 100 epochs with a generalized bell shape MF—‘Gbellmf’ at [3,3,3,3,3]. In the second phase, again the model was trained on a pre-possessed dataset as per the scoring provided in Table 2. From the different experimental tests, the Gbellmf at MF value [3,3,3,3,3] gave the best result with minimum training RMSE: 0.00002, testing RMSE: 0.01167, and checking RMSE: 0.00643 at 100 epochs. The number of data samples for this training are sufficient as well as the obtained RMSE value indicates the best fitting model [61]. The ANFIS structure, defined with a MF value has 524 nodes and 243 rules, generated as the best result, as shown in Figure 5a. Figure 5b shows the error plot during training at 100 epochs. Figure 5c,d display the FIS output with test and validation data during training, in which the predicted output is almost overlapped with FIS output showing the best results of the model. Figure 5e presents the trained ANFIS model at 100 epochs having five input variables and one output (status) variable, while Figure 5f shows the membership function plot of input variable ‘T’, where we consider the threshold of 0.5 for decision making based on the ANFIS output. Figure 5g shows the model input and output fuzzy system. Input at L = 4, S = 2, M = 2, T = 2, and C = 3 gave a status output of 0.997 (almost close to 1, i.e., greater than 0.5), which indicates device upgrades.

The three-dimensional surface view provides the patterns of two input parameters with respect to the corresponding output. For example, the contour view plot in Figure 6a shows that the output status gives a maximum value of 0.04 even if the reference input of S and L both are at the highest value of 4. This implies the device replacement based on constraints defined in Equation (Equation 8). Similarly, the same scenario can be seen in Figure 6b about the distribution plot of T and L for given reference input at S, M, C = [4, −16, 2]. In Figure 6c, the reference inputs are all positive values, i.e., L, S, C = [2, 4, 4]. Based on these reference inputs, we can see the distribution of M and T constitute to status values varying from 0.5 to 1 with an error tolerance at a positive distribution area from 2 to 4, indicating that the device can be upgraded.

### 4.2. Comparative Analysis with Other Classification Methods

For the cross verification of the model performance, we run the experiments with other recent classification methods. The proposed model has only five input variables that are suitable for hybrid learning in ANFIS. The performance of the model is evaluated in terms of error parameters. Other algorithms, for example, linear regression, fine tree, optimizable SVM, linear SVM, and ensemble tree (boosted and optimizable) were used for evaluation. The experiment was run on Windows 10 with an Intel core i7 (8-core) processor, 8 GB RAM, and MATLAB2020a. We consider 30 iterations for training using other methods.

Figure 7 shows the regression plot of an observed vs. predicted result with check data for four methods viz. ANFIS, linear regression, linear SVM, and fine tree. Figure 7a is based on the output classification using binary result defined by Equations (Equation 7) and (Equation 8). To cross verify the mathematical model proposed, Figure 7b–d are plotted based on the output classification defined by Equation (Equation 6). Almost all plots provided the best fitting result between observed and expected.

During the training, testing, and validation of ANFIS, the values of various error and performance metrics are shown in Table 3. RMSE is the least coefficient of determination ($R^{2}$), correlation is almost 100%, and mean absolute error (MAE) is also the least for ANFIS. The standard deviation values for training, testing, and checking data are also not significantly different. This is the model that fits ANFIS the best. Similarly, Table 4 shows the comparative results of error values with other different classification methods. Other methods are evaluated at 10-fold cross validations and 30 iterations. Regarding the computation complexity of ANFIS, the complexity increases with the increase of input variables, while it has good performance for up to five variables [62]. Our proposed model has five input variables and hence, the computational complexity analysis is not significant. We can see the least value obtained using ANFIS in our evaluation except the time of training, which is comparatively higher than other classification methods; since, training time is a one-time task, which is less significant than other performance parameters.

### 4.3. Model Implementations

We have trained, tested, and validated the proposed model through experimentation considering the dataset of a certain series CISCO routers and additionally generated dataset via augmentation as mentioned in Section 4.1. In this section, we implement our suggested model with two standard IP network topologies (UNINET and CERNET) retrieved from the internet topology zoo (http://www.topology-zoo.org/dataset.html, accessed on 12 April 2020) for additional verification as reference implementation.

The work is implemented via a complete simulation using Python programming language, while the output of router status is visualized using Python NetwokX and Matplotlib module. The system simulation environment is as shown in Figure 8. Network graphs in the form of CSV or GML files are loaded into the system based on the simulation environment settings. The device SNMP details are saved in JSON format, and the data is mapped to each router at random during the network’s initial configuration. Once the network has been started, in the running environment, the information collected from DFS is mapped with the information obtained from the SNMP agent to build an input dataset of IP routers to fetch into the trained ANFIS model for classification. The input data mapper and pre-processor module maps the performance parameter values obtained using SNMP with DFS output and transforms the data into standard scoring systems based on criteria defined in Table 2. The dataset generated for each router is fetched into the ANFIS model through Matlab engine API in Python. Matlab engine API provides the decision obtained from the trained ANFIS model and the network with status is visualized as shown in Figure 9, while Figure 9a is the status plot of CERNET and Figure 9b is the status plot of UNINET. Nodes with a red color legend should be replaced, whereas nodes with a green color legend should be updated. Unclassified nodes have a blue color legend. “Unclassified” response indicates the error of the trained model. In these implementations, the ANFIS model is able to classify all the routers in the network. This model shall be used to determine the state of routers in the ideal path for phase-wise migration planning based on the shortest path and customer priority [13].

Figure 9 indicates that there are number of routers classified into upgrade and replacement. Those devices, are supposed to be already SoDIP6 capable, if any are already filtered by DFS as “Do Nothing” and considered “Unclassified” as indicated in Figure 9. The ANFIS module does not consider any device to be classified as non-upgradable or kept as it is, because our assumption is of technology transformations from legacy IPv4 to SoDIP6 capability, while this “non-upgradability” classification is applicable only for regular upgrade/maintenance planning of routers concerning its performance upgrades of its firmware/hardware on the same technology. ISPs can not upgrade or replace all those devices at once due to a higher cost, HR issues, and many more. Following the incremental deployment approach based on optimal path and customer priority, we introduced the proposed ANFIS model to detect the device migration status instead of random assignment via simulation in our previous work [13,22] to achieve more realistic results.

### 4.4. Discussion and Future Work

The first step for every ISPs and Telcos, who are planning to migrate their legacy network into aSDN-enabled IPv6 network is to identify the current operational status of every network devices and identify whether the current networking devices can be upgradeable or requires replacement with latest devices that support targeted newer technologies.

We presented basic steps and a decision making guideline to be taken while migrating the legacy IPv4 networks into SoDIP6 networks with migration cost estimation and optimization in our previous works [13,17] via simulations and analysis. This work implemented the machine learning algorithm to classify the network device with possibility towards upgrades or replacement.

Following the related work discussed in Section 2, Section 3 of this article discussed our proposed concept with methodological framework. In Section 4, we presented the experimental design, training, testing, and validation of our proposed model, while Section 4.3 demonstrated the additional verification of our developed model by implementing it on standard IP routing networks.

Considering the scope, the major challenges of this research are as follows. We are particularly confined to a certain series of CISCO IP routers despite the fact that there are numerous network vendors globally who produce network equipment. This is owed to other vendors’ inability to provide sufficient datasets publicly for analysis. The preprocessing of the data samples by applying scoring and weight provisioning resulted to best test RMSE and other parameter values as shown in Table 3, which is acceptable in ANFIS modeling [61]. The input variables were chosen only considering the SoDIP6 network migration perspectives. The other parameters, such as maintenance ratio, downtime ratio, power consumption status, and many more are sensitive parameters to be considered in device classification in the regular maintenance and upgrade plan. Due to heterogeneous device characteristics, the trained model can not be generalized, while its implementation is specific to CISCO products. The DFS module is conceived based on device support in SDN and IPv6. Hence, the proposed model is particularly applicable to SoDIP6 network migration. Various other variables could be considered for other technology transformations.

Generally, ANFIS has a higher computational cost for a large number of inputs having more than five input variables [62,63]. However there are only five input variables designed in our model. Hence, the proposed approach is not computation/resource intensive. Since, technical administrator monitors the system and device operation continuously using different professional monitoring software e.g., CACTI, PRTG, and Nagios, etc. For a large number of devices in a large network, functionality inspection of individual devices is a complex process. Assessment of network devices using our proposed model helps to take further decision for suitable migration planning of an existing legacy network into the SoDIP6 network. We expect that this approach can easily be integrated into the existing monitoring system and evaluate device performance for its migration assessment. For the enterprise and data center networking, where a large number of switches are in used and they require migration, it is encouraging to implement such an approach in switch migrating planning.

## 5. Conclusions

Service provider network migration to the latest networking is the need of all stakeholders to avoid all the issues in existing network operation and management for future sustainability. The emerging latest generation networking paradigms e.g., IPv6 and SDN should be considered for legacy IPv4 network migration. Switches and routers are the main networking components that must be kept up to date in order to provide customers with effective and reliable services. Large ISPs with thousands of devices in their service network have to consider a need of intelligent approach for cost effective migration planning. In this article, we considered ISP network transformations to the SoDIP6 network to classify the network routing devices in terms of upgrade or replacement by implementing ANFIS. The proposed model outperforms well as compared with other classification methods in terms of performance and accuracy. Additionally, the presented model is verified by implementing with a standard IP network, showing the highest accuracy in network router status identification.

## Figures and Tables

**Figure 1 sensors-22-00143-f001:**
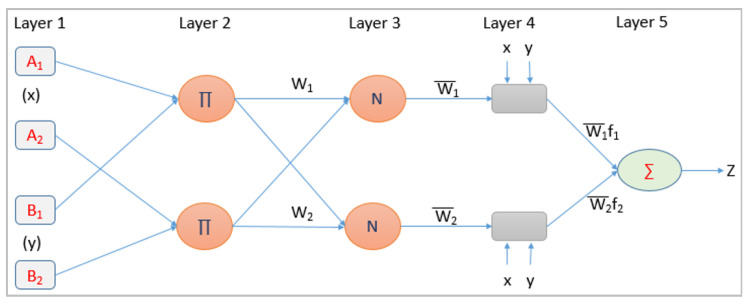
ANFIS structure with two inputs (x, y) and one output (z). ANFIS, adaptive neuro fuzzy inference system.

**Figure 2 sensors-22-00143-f002:**
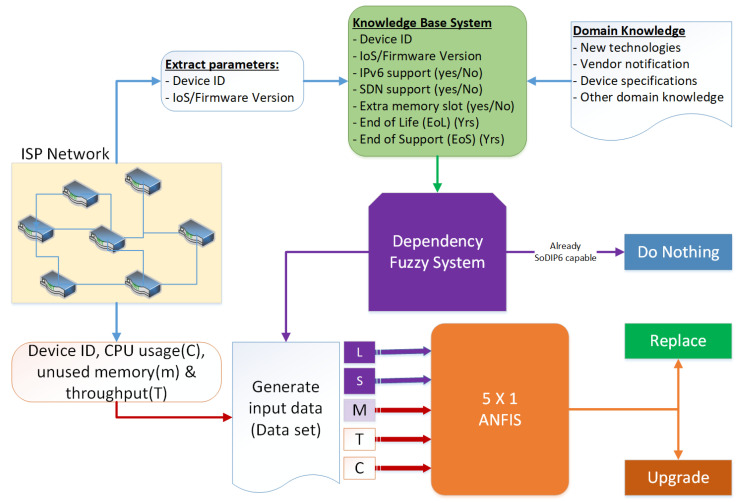
Device status identification implementation framework.

**Figure 3 sensors-22-00143-f003:**
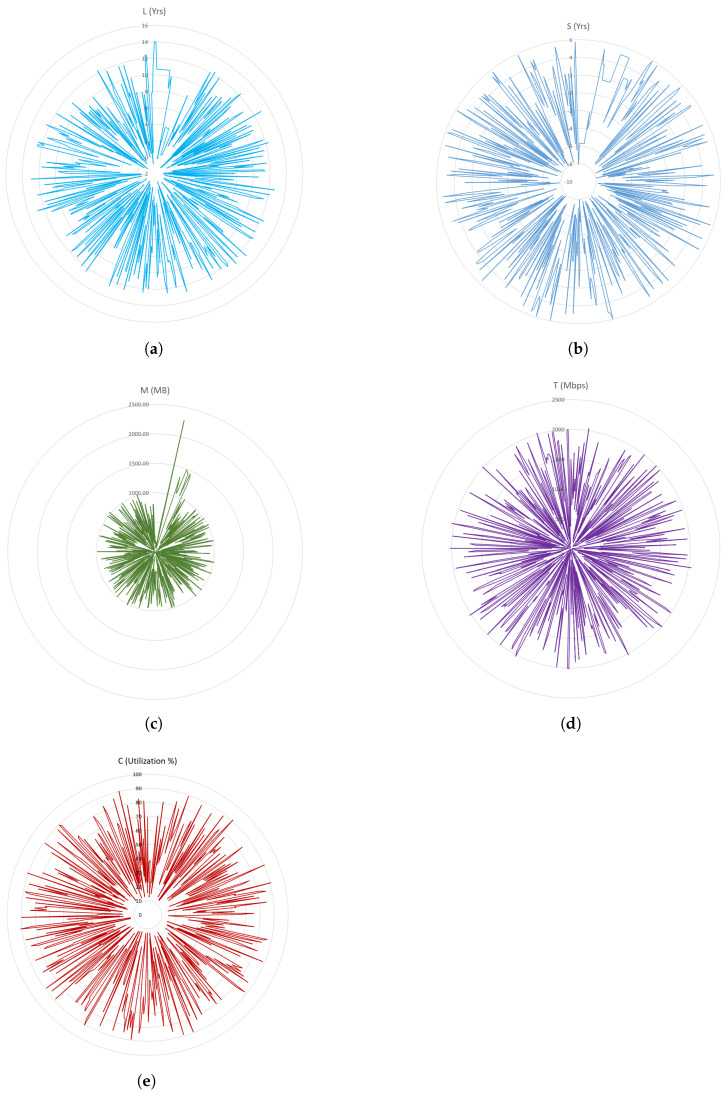
Distribution range of sample dataset (500 samples) of (**a**) End of Life—L, (**b**) End of Support—S, (**c**) Memory—M, (**d**) Throughput—T, and (**e**) CPU utilization—C.

**Figure 4 sensors-22-00143-f004:**
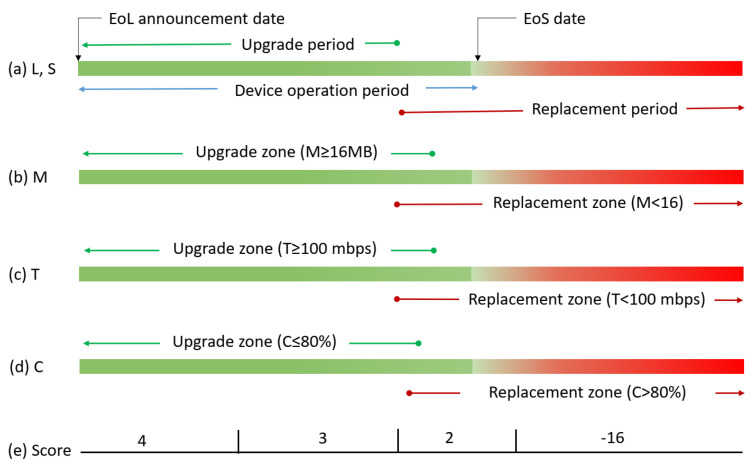
Individual parameter value mapping to score for an upgrade or replacement.

**Figure 5 sensors-22-00143-f005:**
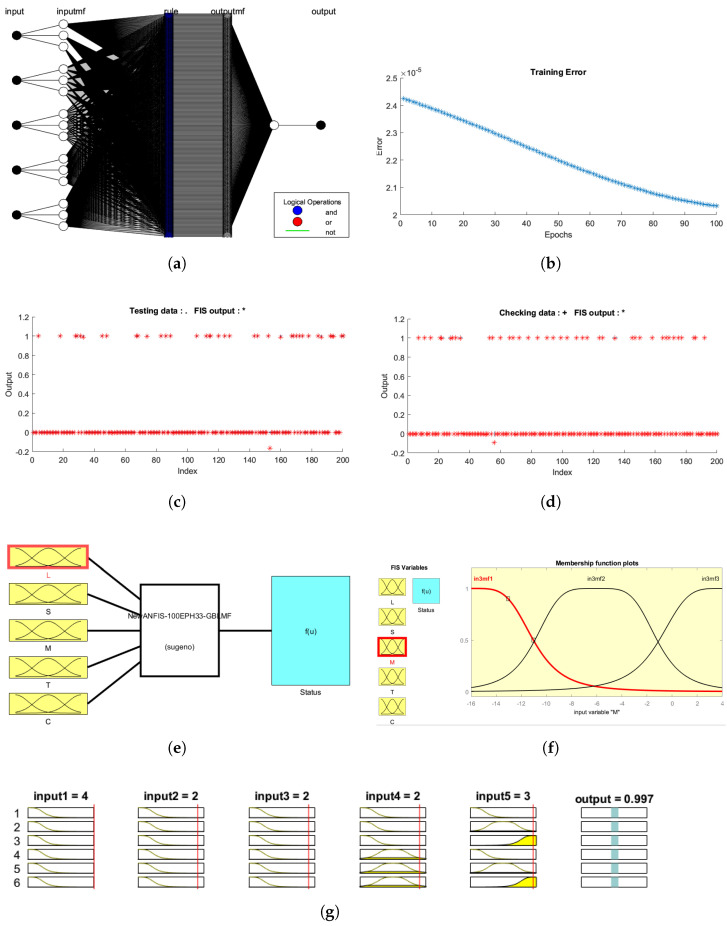
ANFIS model training, testing, and validation. (**a**) Modeled ANFIS (243 rules, 524 nodes). (**b**) Training error stabilized at RMSE = 0.000020. (**c**) Test data and FIS output plot. (**d**) Check data and FIS output plot. (**e**) 5 input and 1 output ANFIS at 100 epochs. (**f**) Membership function plot with variable ‘T’. (**g**) Model output evaluation window at input [L,S,M,T,C] = [4, 2, 2, 2, 3] and output = 0.997.

**Figure 6 sensors-22-00143-f006:**
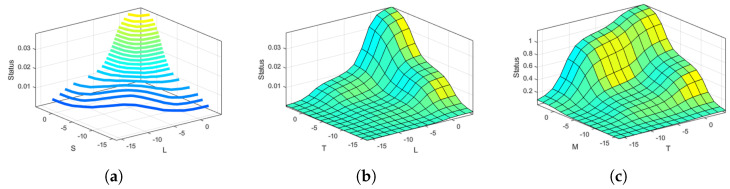
Surface view of input parameters for ANFIS model at different reference inputs. (**a**) Contour view of S and L at ref. inputs: M,T,C = [−16,4,2]. (**b**) Surface view of T and L at ref. input: S,M,C = [4,−16,2]. (**c**) Surface view of M and T at ref. inputs: L,S,C = [2,4,4].

**Figure 7 sensors-22-00143-f007:**
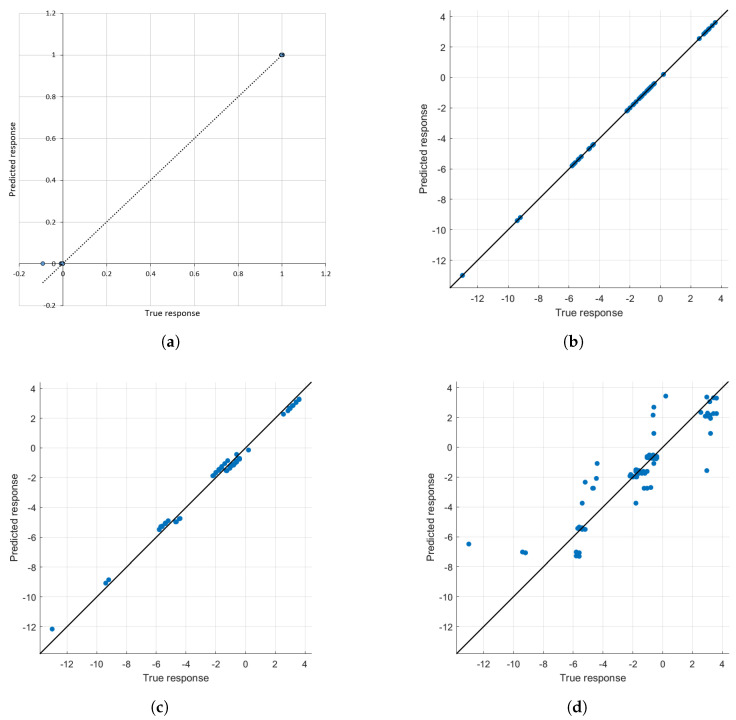
Regression plots by different methods. (**a**) ANFIS classification is based on data samples with threshold defined in Equation (8). (**b**) Linear regression based on data samples with threshold defined in Equation (6). (**c**) Linear SVM based on data samples with the threshold defined in Equation (6}). (**d**) Fine tree regression based on data samples with the threshold defined in Equation (6).

**Figure 8 sensors-22-00143-f008:**
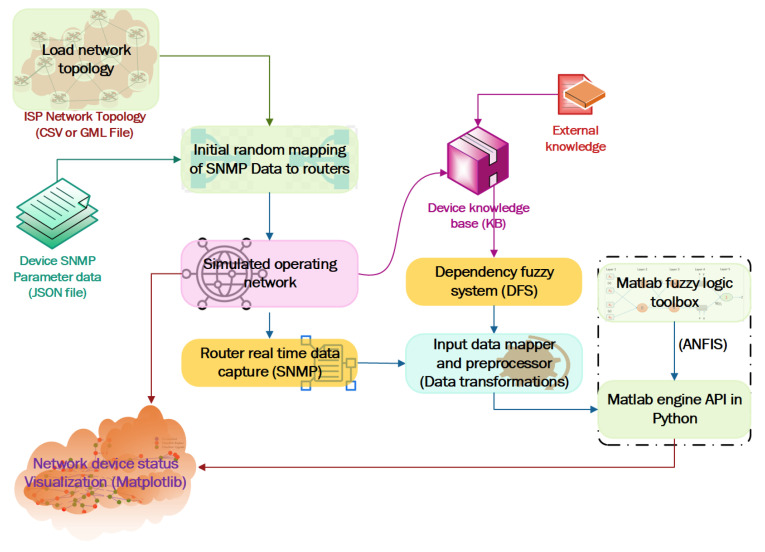
System model simulation environment.

**Figure 9 sensors-22-00143-f009:**
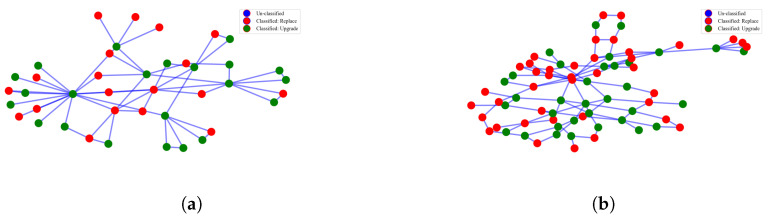
Classification of IP routers by the trained ANFIS model. (**a**) CERNET network. (**b**) UNINET~network.

**Table 1 sensors-22-00143-t001:** Input variables and their semantics [16].

Variable Name	Variable Type	Descriptions
IO	Boolean	IPv6 and SDN/OpenFLow supports
L	Integer	Remaining EoL, date converted into number of years
S	Integer	Remaining EoS, date converted into number of years
m	Integer	Average unused memory of device during operation
$M_{E}$	Integer	Extra memory slot size
M	Integer	Total memory to be after upgrades (=m+$M_{E}$)
T	Integer	Average device throughput
C	Float	Maximum CPU utilization in percentage
E	Integer	Expandable memory size

**Table 2 sensors-22-00143-t002:** Scoring and weight provisioning for data preprocessing [16].

EoL (L-Years)		EoS (S-Years)		Memory (M-MB)		Throughput (T-Mbps)		CPU Usage (C-%)	
Weight = 0.2		Weight = 0.25		Weight = 0.15		Weight = 0.2		Weight = 0.2	
**Value Range**	**Score**	**Value Range**	**Score**	**Value Range**	**Score**	**Value Range**	**Score**	**Value Range**	**Score**
L < 1	4	S < 1	−16	M < 16	−16	T < 100	−16	C < 40	4
1 ≤ L ≤ 2	3	1 ≤ S ≤ 2	2	16 ≤ M ≤ 48	2	100 ≤ T ≤ 1k	2	40 ≤ C ≤ 60	3
2 < L ≤ 4	2	2 < S ≤ 5	3	48 < M ≤128	3	1k ≤ T ≤ 5k	3	60 < C ≤ 80	2
L > 4	−16	S > 5	4	M > 128	4	T > 5k	4	C > 80	−16

**Table 3 sensors-22-00143-t003:** ANFIS performance evaluation with training, testing, and checking data [16].

	RMSE	$R^{2}$	Correlation	Standard Deviation	MAE
Training	0.00002	0.99999	0.99999	0.30503	0.0000044
Testing	0.01167	0.99912	0.99956	0.39254	0.0021872
Checking	0.00643	0.99974	0.99987	0.40029	0.0012512

**Table 4 sensors-22-00143-t004:** Comparison of training error measurement with different classification methods.

	Linear Regression	Fine Tree	Optimizable SVM	Linear SVM	Ensemble Tree (Boosted)	Ensemble Tree (Optimizable)	ANFIS
RMSE	0.22953	0.05244	0.27915	0.26639	0.06098	0.00081	0.0000203
$R^{2}$	0.44000	0.97000	0.17000	0.24000	0.96000	1.00000	0.9999999
MAE	0.18071	0.00364	0.26297	0.15123	0.01270	0.00026	0.0000044
Training time (s)	1.73070	1.59820	436.210	1.66740	2.37680	72.63100	>600.0

## Data Availability

Not applicable.

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
