# Peer review of "Intelligent Approach to Network Device Migration Planning towards Software-Defined IPv6 Networksâ€"

_sensors, 2021, doi:10.3390/s22010143_

Round 1
Reviewer 1 Report
- Full name of IoS should be given the first time it appears in the abstract. The same problem exists with some other abbreviations.
- The subfigures of Figure 3 overlap with each other. Please correct it. There's same problem with Figure 5.
- Recommend authors add a paragraph or two to clarify the improvements made in the journal version based on their conference paper. It helps readers see the difference when referencing their work.
"Dawadi, Babu R., Danda B. Rawat, Shashidhar R. Joshi, and Pietro Manzoni. "ANFIS based Classification Model for Network Device Migration towards SoDIP6 Networks." In 2021 IEEE 18th Annual Consumer Communications & Networking Conference (CCNC), pp. 1-6. IEEE, 2021."
Author Response
Respected Reviewer,
We are thankful to reviewer for your careful reviews and time. We have fully refined the manuscript accordingly, following are our response,
- We refined the manuscripts and abbreviations are provided at first time use.
- The subfigure overlap problems on all figures are fixed
- We have added a header note at be beginning of the article by referring our conference paper (ref. [63]) and the extra works we did in this article. We properly cited our conference paper in the main text too (Ref. [63])
Once again we thank you for time and careful reviews.
Reviewer 2 Report
Internet and telecom service providers require to successfully migrate existing IPv4 legacy networking systems to IPv6 as well including software-defined networking (SDN), which is an emerging paradigm. Bench-marking of existing networking devices is therefore required to identify their current functionality and determine whether they can be easily upgraded or will need replacement to make them fit for purpose. This paper uses an adaptive neuro fuzzy inference system (ANFIS) to provide an intelligent way of determining network device status identification by classifying whether a network device is upgradable or requires replacement. A knowledge-base is also proposed to store metadata on device IoS/firmware version, SDN and IPv6 support requirements, together with end-of-life and end-of-support dates. For input to ANFIS, device performance metrics such as average CPU utilization, throughput, and memory capacity are stored and mapped to other data held in the knowledge-base. The ANFIS approached compared with other well-known classification methods, such as support vector machines and shown to compare favourably in terms of accuracy and other common metrics.
This is an interesting approach which is likely to prove useful for addressing such problems of implementation, migration and ensuring compatibility of changing IP standards. The problem is well articulated and clearly described. The contributions and novelty are also clearly presented in a well-structured manner. In addition, the approach is well motivated and nicely presented, as is the experimental design, experimentation and choice of data sets which were generated from 40 different CISCO product models. Finally, the results are clearly set out and thoroughly evaluated, as well as the further work being clearly detailed.
Overall, this is a very useful piece of work which is very well motivated and clearly presented. The methods, experimentation, results and their interpretation are very clearly described and discussed at an appropriate level of detail. In my opinion this paper is well-deserving of publication.
Author Response
Respected Reviewer,
We have highly thankful for your careful and thorough reading of the manuscript and provided very nice reviews/summary of our work.
Thank you for your time and reviews with publication recommendation.
Regards
Authors